# The impact of a Post-Take Ward Round Pharmacist on the Risk Score and Enactment of Medication-Related Recommendations

**DOI:** 10.3390/pharmacy8010023

**Published:** 2020-02-20

**Authors:** Brooke Bullock, Peter J Donovan, Charles Mitchell, Jennifer A Whitty, Ian Coombes

**Affiliations:** 1Pharmacy Department, Royal Brisbane and Women’s Hospital, Herston QLD 4029, Australia; 2School of Pharmacy, University of Queensland, Woolloongabba QLD 4102, Australia; 3Medical Education Unit, Gold Coast Hospital and Health Service, Southport QLD 4215, Australia; 4Faculty of Medicine, University of Queensland, Herston QLD 4006, Australia; 5Department Clinical Pharmacology, Royal Brisbane and Women’s Hospital, Herston QLD 4029, Australia; 6Norwich Medical School, University of East Anglia, Norwich NR4 7TJ, UK

**Keywords:** drug-related problems, clinical significance, review, medication, hospital, prescribing

## Abstract

There is a scarcity of published research describing the impact of a pharmacist on the post-take ward round (PTWR) in addition to ward-based pharmacy services. The aim of this paper was to evaluate the impact of clinical pharmacists’ participation on the PTWR on the risk assessment scores of medication-related recommendations with and without a pharmacist. This includes medication-related recommendations occurring on the PTWR and those recommendations made by the ward-based pharmacist on the inpatient ward. A pre–post intervention study was undertaken that compared the impact of adding a pharmacist to the PTWR compared with ward-based pharmacist services alone. A panel reviewed the risk of not acting on medication recommendations that was made on the PTWR and those recorded by the ward-based pharmacist. The relationship between the risk scores and the number and proportion of recommendations that led to action were compared between study groups. There were more medication-related recommendations on the PTWR in the intervention group when a pharmacist was present. Proportionately fewer were in the ’very high and extreme’ risk category. Although there was no difference in the number of ward pharmacist recommendations between groups, there was a significantly higher proportion of ward pharmacist recommendations in the “very high and extreme” category in those patients who had been seen on a PTWR attended by a pharmacist than when a pharmacist was not present. There were a greater proportion of “low and medium” risk actionable medication recommendations actioned on the PTWR in the intervention group; and no difference in the risk scores in ward pharmacist recommendations actioned between groups. Overall, the proportion of recommendations that were actioned was higher for those made on the PTWR compared with the ward. The addition of a pharmacist to the PTWR resulted in an increase in low, medium, and high risk recommendations on the PTWR, more very high and extreme risk recommendations made by the ward-based pharmacist, plus an increased number of recommendations being actioned during the patients’ admission.

## 1. Introduction

Drug-related problems (DRP), which are associated with increased morbidity, mortality, and healthcare costs, must be identified and addressed [1]. This is a current global health challenge [2]. There are numerous ways in which hospitals aim to reduce DRPs and optimize medication use for patients, one of which are clinical pharmacists. The literature provides strong evidence for the impact of clinical pharmacists, who work with medical officers to optimize the use of medicines and improve the safety and appropriateness of prescribing [3]; they assist with the identification, resolution, and prevention of DRPs reducing morbidity, hospital mortality rates, and medicine-related incidents in the majority of cases [4]. In Australia, this medication review by a pharmacist usually occurs on the inpatient ward and, for the purposes of this paper, are termed “ward-based pharmacy services” [5].

An ideal opportunity for identification and resolution of DRPs is on the ward round where the medical officers and pharmacists have been shown to discuss and optimize patient’s medication management [6]. This paper focuses on a specific type of ward round, called the post-take ward round (PTWR) where the admitted patient is first presented to the treating consultant medical officer and an initial inpatient management plan is formulated [5,7]. It has been demonstrated that clinical pharmacist attendance on the PTWR leads to an increase in the number of medication-related discussions with more appropriate medications prescribed at the time of discharge [5,7].

The Society of Hospital Pharmacists of Australia (SHPA) recommends that a risk assessment should be done for clinical interventions regarding medications in order to assess the potential significance of their impact and assist prioritization of resources [3]. Further to what we know about pharmacists working on PTWRs, this paper investigates the impact of a PTWR pharmacist on the level of risk assigned to all medication-related recommendations both on the PTWR and subsequently on the ward: This has not been previously reported. Risk scoring often includes a description of the adverse consequences and their likelihood if there is no intervention [8]. 

## 2. Aim

The aim of this study is to evaluate the impact of clinical pharmacist participation on the PTWR on the risk assessment and enactment of medication-related recommendations discussed on the PTWR and subsequent recommendations made by the ward-based pharmacist.

## 3. Ethics Approval

Ethics approval was obtained from the Hospital and University Human Research Ethics Committees (HREC/13/QRBW/443;2014000705). Verbal consent was obtained and recorded for all patients and written consent was obtained from all pharmacists and medical officers (MOs) prior to the period of observation. Consent could be withdrawn at any time.

## 4. Methods

### 4.1. Study Design:

The study occurred in the Internal Medicine Department at a 929-bed quaternary and tertiary referral teaching hospital in Brisbane, Australia. In brief, in the initial six-week phase (comparator group: April to June 2014), pharmacist input occurred on the ward to which the patient was transferred after the PTWR with no pharmacist present on the PTWR. By contrast, in the second six-week phase (intervention group: August to October 2014), one of four senior clinical pharmacists were present on the PTWR and interacted with medical staff: This was in addition to ward-based pharmacy input. One experienced pharmacist observed PTWRs in both study phases but did not participate in any prescribing decisions unless there was a need to intervene to prevent or highlight a potential adverse event that had so far not been recognized and that may have caused immediate harm. The presence of the PTWR in the intervention group was in addition to the usual ward-based pharmacy service, which was present in both groups.

During both study phases, all observed medication-related communication on the PTWR was recorded. Retrospectively, all medication-related recommendations made by the ward pharmacist and recorded on the Medication Action Plan (MAP) were recorded.

The methods for the comparator and intervention observation study are described in further detail in an earlier publication by Bullock et al. [8].

### 4.2. Participants and Data Collection

Patients (18 years or older) under the care of all eight of the hospital’s internal medical teams were observed in both study phases. Inpatient medication order forms, the Discharge Medication Record (DMR), and medical records were used to collect data before the PTWR and on hospital discharge. One experienced pharmacist observed discussions related to individual patients for both periods using a structured form to collect patient and medication details communicated.

All medication-related communication was recorded. Any two-way discussion between medical staff or the pharmacist as to whether a treatment modification was described.

The above briefly summarizes the comparator and intervention observation study described in an earlier publication by Bullock et al. [5]. Below is a description of the risk assessment of medication recommendations and their impact.

### 4.3. Risk Scoring of Medication-related Recommendations 

A panel of two senior pharmacists and a senior medical officer (an advanced trainee clinical pharmacology registrar), all blinded to the arm of the study, completed the risk scoring of all medication-related communication for the comparator and intervention cohorts for both: (1)Medication-related recommendations that occurred on the PTWR, and;(2)Ward-based clinical pharmacist recommendations recorded on the MAP.

To complete the risk scoring, each episode of medication communication was discussed by the panel, along with relevant patient clinical information until a consensus was reached.

Risk scoring was completed using the PRIME risk matrix (see Appendix A, Table A1) to describe the probability of an adverse outcome occurring had a recommendation not been enacted, using a 5-point scale (see Appendix A, Table A2) from rarely (may occur in exceptional circumstances) to almost certain (is expected to occur frequently). As well, the consequence of that event using a 5-point scale (see Appendix A, Table A2) of negligible (where no injury or harm would be caused) to catastrophic (with the likely consequence of death) was estimated. Following this, the probability and consequence were used to determine an overall risk score categorized into five levels of low, medium, high, very high, or extreme, which then we grouped as “low and medium”, “high”, or “very high and extreme”. 

### 4.4. Enactment of Medication-related Recommendations

For all medication-related recommendation on the PTWR and the ward, proposed treatment modifications were noted as was whether they were able to be actioned (e.g., clear, measurable modifications were suggested) and if they were actioned at any time during the patients stay (the proposed change was recorded on the patients inpatient medication order form/s, DMR, and/or documented in the patients’ medical record).

### 4.5. Outcome Measures

The difference between the comparator and intervention groups were compared for:The “low and medium”, “high”, and “very high and extreme” risk scores,The proportion of recommendations that were actioned.

These comparisons were carried out for both the medication-related recommendations that occurred on the PTWR with and without a pharmacist and the ward-based recommendations made by pharmacists subsequent to PTWR.

### 4.6. Data Analysis 

R commander version 3.2.4 (2016-03-10) was used for data analysis. Patient demographics and other continuous data are presented as mean ŷ standard deviation (or median and inter quartile range where appropriate). Chi-squared tests were used to compare differences in proportions. T-tests were used to compare differences in means where data were normally distributed. Categorical and binary data have been expressed as counts and percentages of the total number of possible outcomes. Statistical significance was achieved if *p* < 0.05.

## 5. Results

### 5.1. The Clinical Significance (Risk Scoring) of Medication Recommendations on the PTWR 

The panel was able to risk rate 230 of the 249 (92.4%) medication-related discussions in the comparator group and 326 of the 352 (92.6%) in the intervention group as well as 170 of the 195 (87.2%) of the ward pharmacist recommendations documented on the MAP in the comparator group and 168 of the 225 (74.7%) in the intervention group. The remainder of the medication recommendations were unable to be risk rated due to a lack of documented information and were excluded from the following analysis. Examples of the risk scoring of medication-related recommendations are shown in Table 1.

Table 2 details the medication recommendations across comparator (no pharmacist present on the PTWR) and intervention groups (Pharmacist present and inputting to the PTWR) according to their risk score. This table is presented in two parts: Part 1 is PTWR recommendations for both the comparator and intervention groups. Part 2 is the ward pharmacist recommendations, which occurred on the inpatient wards: These occurred after the PTWT when the patient was admitted to the ward. 

### 5.2. Number of Medication-related Reommendations 

There was a significant increase in the number of PTWR medication-related recommendations (comparator 249, baseline 352, *p* < 0.001) with the presence of a PTWR pharmacist, which was due to a significant increase in the number of “low and medium” and “high” risk medication-related recommendations, and a similar number of those rated as “very high and extreme” risk.

By contrast, there was no change in the total number of ward pharmacist recommendations in patients seen when a PTWR pharmacist was present (comparator 195, intervention 225, *p* = 0.22). When comparing those able to be risk rated however, more of their recommendations had a “very high and extreme” risk score—32 compared with 13 in the comparator group.

### 5.3. Proportion of Medication-related Recommendations that Were Actioned

Overall, there were significantly more of the actionable PTWR medication-related recommendations actioned in the intervention group where there was a pharmacist present on the PTWR (comparator 167 of 216 (77%), intervention 264 of 285 (86%), *p* = 0.004). There was a significant increase in the proportion of medication-related PTWR recommendations actioned that were rated as “low and medium” risk and no significant changes in the other two risk groups. 

Overall, there were a similar number of ward pharmacist recommendations actioned for both groups (comparator 120 of 170 (71%), intervention group 116 of 168 (69%), *p* = 0.76). The proportion of ward pharmacist recommendations recorded on the MAP that were actioned across all levels of risk score remained similar across risk groups.

### 5.4. Proportion Actioned Depending on Setting 

Overall, 85.2% (167/196) of the PTWR medication-related recommendations were actioned in the comparator group without any pharmacy input and 93.7% (264/285) in the intervention group. This was significantly greater than the proportion of those recommendations actioned by the ward pharmacist regardless of if a pharmacist was present on the PTWRD, with 70.6% (120/170) actioned in the comparator group and 69% (116/168) in the intervention group.

## 6. Discussion

This paper assigns a panel derived risk score to medication-related communication on the PTWR, with and without a pharmacist input and the subsequent recommendations made by the ward-based pharmacist (as documented on the MAP); as a measure of the significance of the contribution made. This serves to provide previously unreported information on the risk scoring of medication communication in these specific settings and explores how those scores are influenced by the addition of a PTWR pharmacist. This can assist with prioritizing the role of clinical pharmacists. 

In the earlier publication by Bullock et al. [5] describing medication appropriateness outcomes for the same study groups being investigated in this paper, each medication prescribed on the PTWR was rated as “high risk” or “not high risk” according to the Australian Safety & Quality Council’s APINCH classification system [9]. Although this system of risk assignment is useful and simple to implement, a better way to determine the clinical significance of each medication discussion and the recommendations made by the pharmacist on the PTWR and the ward-based pharmacist should include patient factors such as diagnosis, past medical history, comorbidities, as well as drug-related factors including other medication, rather than the actual medication class discussed. The results of this study highlight that although there is benefit in prioritizing medication review based on a dichotomized categorization system such as APINCH, pharmacists should consider an assessment of the probability and potential consequence of addressing the potential or actual DRP as a result of each medication in the context of the individual patient. 

Overall, as expected, the presence of a pharmacist on the PTWR resulted in significantly greater number of medication-related issues being identified and discussed with medical staff. In particular, there were more low, medium, and high risk medication-related recommendations and a similar number of recommendations rated as “very high and extreme” risk; there was a non-significantly greater proportion actioned. Additionally, there was a higher proportion of medication recommendations actioned in those scored as low and medium risk in the intervention group. 

The reason for this result may be that the presence of the pharmacist, in collaboration with the senior medical officers on the ward round, stimulated a focus on the patient’s medication and highlighted the importance of low, medium, and high risk interventions to patients’ overall medication appropriateness and ensured a greater proportion were actioned by the team during the patients stay. 

There was a similar overall number of ward pharmacist recommendations between comparator and intervention groups, however a significant increase was noted in those recommendations of very high and extreme risk between in the intervention group. These results suggest that the PTWR pharmacist may have an impact over and above what is achieved while on the PTWR, by, enabling the ward pharmacist to prioritise more high value interventions. These results combined, may help explain the improvements in medication appropriateness shown in the results previously published for these study groups [5].

Despite the improvement between comparator and intervention groups, there were a number of recommendations and recommendations not actioned in both study groups. This highlights an opportunity for follow up of the recommendations made on both the PTWR and ward, with a priority made to follow up on those recommendations of higher clinical significance. A verbal and written handover (for example, via the MAP) to the ward pharmacist and team to ensure follow up is essential, which was not mandated in the model of care investigated in this study. The same pharmacist fulfilling both PTWR and ward roles may be an alternative model to enable follow through on the identified issues however comes with greater demand on staffing and skill mix within a hospital pharmacy department.

This study did not investigate the cost or cost savings of the intervention. This paper, along with the earlier publication by Bullock et al., provides further evidence of improved medication communication and appropriateness with such an intervention. This evidence calls for further economic evaluation to assess whether a PTWR pharmacist service is cost-effective and possibly cost saving to the health system [5]. To provide a grounding for future work in this area, Bullock et al. recently published a systematic review of the costs and cost-effectiveness of clinical pharmacists on hospital ward rounds, which found seven studies of a clinical pharmacist’s inclusion on hospital ward rounds; none were deemed to be a full economic evaluation [10].

Many pharmacy services prioritise services to inpatient ward-based review over pharmacist attendance on ward rounds to ensure completion of key medication management interventions that are mandated in National safety and Quality standards such as best practice medication history, medication reconciliation on admission, and discharge education and reconciliation in order to ensure continuity of quality use of medicines and enhanced patient care [11].This study illustrates a higher proportion of recommendations that were actioned on the PTWR as opposed to the ward-based service alone. It is worth noting that, as recommended by previous research in this area [12], the PTWR pharmacists in this study had well-developed clinical knowledge, communication, and interpersonal skills and were able to actively engage with the senior medical decision makers on the ward round as opposed to the more junior ward-based pharmacists working with the predominantly junior medical staff. 

### Limitations: 

The most relevant limitation to this paper was that there was no system in place to ensure a formal written handover between the PTWR pharmacist and ward pharmacist. Some handover occurred through written documentation on the National Inpatient Medication Chart or the Medication Action Plan, or verbal handover between pharmacists. However, there was no formal handover for the majority of patients.

As mentioned earlier in the paper, the comparator and intervention observations were not carried out at the same time of the year (the comparator phase between April and June 2014, and intervention phase between August and October 2014) and therefore other changes within the hospital such as experience of junior medical and pharmacy staff may have impacted outcomes. The observation periods, however, were chosen to minimize this limitation. In addition, the lead researcher was not completely independent. Observation bias was minimized by having the same observer recording data for both the comparator and intervention groups using a standardized data collection method and tool. 

There was some duplication of recommendations on the PTWR and by the ward-based pharmacists. These recommendations have not been excluded as the nature and extent of the duplication differs greatly and, hence, including but stating the duplicated numbers was the most appropriate action. The limitations for this comparator and intervention study are described in more detail in the earlier publication by Bullock et al [5].

## 7. Conclusions

The inclusion of a pharmacist on the PTWR is associated with more medication-related recommendations than with the medical team alone, with a greater proportion rated as “low and medium” and “high” and a higher proportion leading to action in those recommendations scored as “low and medium”. In addition, ward-based pharmacists seeing patients after a pharmacist attended PTWR made more recommendations of “very high and extreme” risk medication than when the PTWR did not include a pharmacist. Where resources allow the addition of pharmacists to the PTWR in addition to prioritized ward-based services, this is likely to improve medication appropriateness over and above that of existing pharmacy services. 

## Figures and Tables

**Table 1 pharmacy-08-00023-t001:** Examples of the risk scoring system.

Overall Level of Risk	Example	Probability	Consequence	Overall Risk
Low	Medication access/Substitution - patient did not bring own supply of rosuvastatin which is not available at hospital. Recommendation made to consider withholding during hospital admission or change to atorvastatin.	Unlikely	Negligible	Low
Medium	Medication dosage: Patients prescribed thyroxine dose usually 100 mcg on Sat-Sun, 50 mcg Mon-Fri. Currently charted 50 mcg daily. Recommendation to review and chart as appropriate.	Possible	Minor	Medium
High	Medication withdrawal: Patient charted for regular benzodiazepam. Recommendation made to review need on discharge with view of dementia and increased risk of falls.	Possible	Moderate	High
Very high	Medication cessation: Patient charted for atenolol and sotalol. However, medication history revealed that atenolol was changed to sotalol 2 months prior to hospital admission. Recommendation to cease the atenolol and continue sotalol alone.	Almost certain	Moderate	Very high
Extreme	Drug interaction: Patient prescribed enoxaparin and warfarin therapy. However, INR has 3.9. Recommendation to cease enoxaparin and withhold warfarin until INR is within target range.	Almost certain	Major	Extreme

**Table 2 pharmacy-08-00023-t002:** The risk scores and enactment of medication-related medication recommendations on the post-take ward round (PTWR) and those made by ward-based pharmacists.

	Comparator: No Pharmacist Present on PTWR N = 130 Patients	Intervention: Pharmacist Present on PTWR N = 130 Patients	*p* Value
**Part 1—Post-Take Ward Round - medication related recommendations**			
**Low & medium**			
Number of medication related recommendations (mean number per patient ± standard deviation)	90 (0.69 ± 0.88)	142 (1.09 ± 1.15)	0.002*^*
Proportion actioned *(number actioned/number able to be actioned)*	55/71 78%	112/121 93%	0.003*
**High**			
Number of medication related recommendations (mean number per patient ± standard deviation)	70 (0.54 ± 0.80)	115 (0.88 ± 0.91)	0.001*^*
Proportion actioned *(number actioned/number able to be actioned)*	59/65 91%	96/106 91%	0.965*
**Very high and extreme**			
Number of medication related recommendations (mean number per patient ± standard deviation)	70 (0.54 ± 0.75)	69 (0.53 ± 0.78)	0.935*^*
Proportion actioned *(number actioned/number able to be actioned)*	53/60 88%	56/58 97%	0.093*
**Part 2 — Ward-based – Ward Pharmacist Recommendations documented on Medication Action Plan**			
**Low & medium**			
Number of medication related recommendations (mean number per patient ± standard deviation)	112 (0.86 ± 1.39)	98 (0.75 ± 1.23)	0.509*^*
Proportion actioned *(number actioned/number able to be actioned)*	82/111 74%	64/94 68%	0.361*
**High**			
Number of medication related recommendations (mean number per patient ± standard deviation)	48 (0.37 ± 0.64)	49 (0.38 ± 0.66)	0.924*^*
Proportion actioned *(number actioned/number able to be actioned)*	29/46 63%	30/46 65%	0.828*
**Very high and extreme**			
Number of medication related recommendations (mean number per patient ± standard deviation)	13 (0.10 ± 0.3)	32 (0.25 ± 0.51)	0.006*^*
Proportion actioned *(number actioned/number able to be actioned)*	9/13 69%	22/28 79%	0.517*

Statistical tests: ^ - Welch Two sample t-test, * - Chi-squared.

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
