# Peer review of "The impact of a Post-Take Ward Round Pharmacist on the Risk Score and Enactment of Medication-Related Recommendations"

_pharmacy, 2020, doi:10.3390/pharmacy8010023_

Round 1

Reviewer 1 Report

The manuscript is well written, detailed, and organized. I have the following suggestions/comments:

The background/introduction is very good. I suggest that PTWR and Ward based pharmacy services be defined/described as these terms are not known/used everywhere. Consider adding this to the background/introduction section.

Comment: The results of this study are not surprising, but it does add to the literature since there is a "scarcity of published research" in this area. 

Line 54: suggest to correct the British spelling of "prioritization". Please check the entire manuscript for other such spellings. Thank you.

Please clarify Table 2 - part 1 and part 2 versus the comparator or intervention. Part 2 of the table is confusing. Please explain the difference between the two parts and what is being accomplished. Is there a more straight-forward way to present this information?

Suggestion: With the Methods section (line 73-74) and the Limitations section (line 241-242) - suggest to put these sentences ("described in more detail in an earlier publication"...) at the end of those sections rather than leading off that section with this statement. The sentence (when placed at the end) could read something like... For more information, see the earlier publication by Bullock, etc...

The font on page 8 (lines 267 - 287) seems to be larger from the rest of the manuscript. Please check this and correct.

Thank you! 

Reviewer 2 Report

In general this is a very interesting study and obviously shows the great impact that pharmacists have.  I understand that the difference between this study and the previous one described is that in the previous study you were evaluating the number of DRPs and whether medication was properly prescribed and in this study you are looking at the impact of these interventions. However, this paper simply seems like a sub-analysis of the previously conducted study and should have been published together. 

General

1. Please make sure that references and tables follow guidelines 

Introduction

2. Lines 59 - 61 state the aim/objective of the study; you have something similar in the aims section, so I don't think that you need to have both of them. Furthermore, what do you mean when you refer to risk scores of medication related communication? You have this described further in lines 109 - 111 so I would recommend that you include a bit of this in the introduction

3. In general, I feel like you can perhaps extend the introduction to include other studies or data from pharmacist intervention in clinical teams/rounds. You do have information about one other study, however I think it would be beneficial to have other studies with data. 

Ethics Approval

3. Line 69 - I am assuming that MO stands for medical officers, however please make sure to define this as it may not be inherently clear upon first reading 

Methods

4. Line 74 - add a period at the end of the sentence after the citation. 

5. Lines 76 -85 - this is a bit hard to understand upon first read; please consider the addition of a diagram to allow for better readability

Discussion

6. Line 189 - please add a period at the end of the sentence

Reviewer 3 Report

Interesting article on the utilization of clinical pharmacist in Australia.

Major Revisions

1.Build a better case of the importance of the issue in the introduction. The Bullock et al. article referenced does a really good job orienting readers to the role of clinical pharmacist in Australia.  So describe the overarching problem that your research is trying to solve, then what is know about the topic, what knowledge gap are you filling?

2. Would it be beneficial in you could discuss the economic impact of clinical pharmacist.  As the Bullock et al. article states they evaluated the impact of clinical pharmacist participation on the PTWR on medication prescribing as measured by medication appropriateness, the level and extent of
medication communication and patient and service delivery
related outcomes, including length of stay.  While your aim was to evaluate the impact of clinical pharmacist participation on the PTWR on the risk assessment and enactment of medication related communication on the PTWR and subsequent recommendations made by the ward-based pharmacist.

Minor

1. In your methods section, line 77 could you include the dates for the study period.

Round 2

Reviewer 2 Report

thank you